# Challenges Faced in Large-Scale Nucleic Acid Testing during the Sudden Outbreak of the B.1.617.2 (Delta)

**DOI:** 10.3390/ijerph19031573

**Published:** 2022-01-29

**Authors:** Pingting Zhu, Meiyan Qian, Qiwei Wu, Xinyi Liu

**Affiliations:** 1School of Nursing, Yangzhou University, Yangzhou 225009, China; qmy0301@163.com (M.Q.); MZ120201760@yzu.edu.cn (Q.W.); MX120190879@yzu.edu.cn (X.L.); 2Jiangsu Key Laboratory of Zoonosis, 136 Jiangyang Middle Road, Yangzhou 225009, China

**Keywords:** nucleic acid testing, challenges, the B.1.617.2 variant, COVID-19, qualitative research, diagnostic testing

## Abstract

The Delta variant (B.1.617.2) has dominated in many countries over the world. Its sudden outbreak in China has led the government to quickly carry out large-scale nucleic acid testing to curb its spread. This qualitative study aims to find the challenges based on empirical evidence from the perspectives of the different groups of people involved in the testing, and further explore possible strategies to improve the efficiency of large-scale nucleic acid testing. Using a phenomenological approach, we selected 35 participants (seven managers, eight health professionals, six community volunteers and 14 residents) by purposive sampling. The interviews were conducted by in-depth semi-structured interviews and the data were analyzed by Colaizzi’s seven-step method. Qualitative analysis revealed three main themes: unreasonable and unsafe testing points layout settings, human and medical resources challenges, and potential infection risk. From the different angles, participants all experienced challenges during large-scale nucleic acid testing, making positive planning and adequate preparation important parts of the smooth development of testing. Large-scale nucleic acid testing relies on the cooperation and efforts of all to support containment of the spread of the virus. Local governments should improve their ability to respond to and deal with public health emergencies.

## 1. Introduction

Coronavirus disease (COVID-19) has rapidly developed into a pandemic, causing serious public health problems worldwide since it was first reported in December 2019 [1]. As of 7 September 2021, the COVID-19 has caused over 220 million reported cases and 4.5 million deaths globally [2]. Effective infection control should depend on mass and positive diagnostic testing [3]. Large-scale nucleic acid testing can detect infected people early and play a key role in preventing the spread of the epidemic. Meanwhile, through large-scale testing, we can better understand the rules of local epidemic transmission and judge the trend of the epidemic. Large-scale nucleic acid testing has been carried out in many countries [4,5,6,7]. Some studies believe that mass diagnostic testing could end the epidemic [8,9,10]. Some scholars also advocate the use of repeated testing as a COVID-19-locked exit strategy [9].

On 28 July, Yangzhou, a third-tier city in China, reported the first confirmed case in 2021. In a brief period, the outbreak spread rapidly among dense populations in the urban area of Yangzhou, the government immediately organized mass screening for nucleic acids testing. Health professionals were responsible for nucleic acid testing and many volunteers as staff members to carry out on-site guidance, information registration and other work. These outbreak was caused by the B.1.617.2 (Delta) variant of SARS-CoV-2, which was first detected in India in December 2020 [11]. The Delta variant (B.1.617.2), which is dominant in the UK, India, and Indonesia now [12,13,14], has been detected in over 120 countries [15]. Data have shown that the Delta variant (B.1.617.2) is more transmissible, and associated with high hospitalization rates than other variants [16].

Once a localized outbreak is detected, epidemiologists classify geographic jurisdictions into low-risk, medium-risk, and high-risk areas according to the risk of transmitting the virus [17]. As of 12 August 2021, there are three high-risk areas and 84 medium-risk areas in Yangzhou, where community transmission or hidden transmission may exist, and the population in these areas is the focus of “should be tested”. However, it is not easy to fully achieve the goal of detecting all the infected people or the source of infection through a one-time, point-in-time test, repeated testing for all population is performed to detect outbreaks before they spread [18].

Several challenges encountered in large-scale nucleic acid testing have been reported in some studies. First, the limitations of testing lie in the substantial assay time, the high cost of the detection reagent, which is a challenge to the economic strength of a region [19]. Due to a simultaneous increase in the purchases at the global level, materials required for nucleic acid testing are in short supply. Second, it needs a lot of manpower to keep the process going effectively, and the lack of a consistent national strategy to fight the epidemic will also limit the efficiency and scale of the testing [20]. In addition, the collection of nucleic acid specimens requires close contact between health professionals and patients, which lead to a risk of transmission of the virus to the health professionals [21]. Compared to first-tier cities such as Beijing and Guangzhou, third-tier cities are relatively less able to react and respond to public health emergencies. In multiple rounds of nucleic acid testing in Yangzhou, nucleic acid testing point transmission chains have also occurred due to inexperience and defective tissue management. For this reason, the purpose of this study is to describe the challenges that people faced when participating in large-scale nucleic acid testing, which helps to optimize the tissue of the whole large-scale testing, ensure the safety of detectors and subjects, avoid the occurrence of cross-infection. At the same time, it provides valuable experience and direction for better carrying out large-scale nucleic acid testing in the face of the sudden epidemic in other countries.

## 2. Materials and Methods

### 2.1. Study Design

We conducted a qualitative study, phenomenological research was used to describe the perception and challenges that people faced when participating in large-scale nucleic acid testing in Yangzhou. Phenomenological research focuses on describing commonalities of experiences across the entire population. The contents of the interviews were collated and analyzed according to the phenomenological analysis of the Colaizzi.

### 2.2. Particapiants

The participants were chosen from three different testing points, and the participants in all four groups (managers, health professionals, community volunteers, residents) were purposively chosen to allow the capture of all groups’ experience. The inclusion criteria for participants consisted of the following: (1) participated in the nucleic acid testing, (2) the most recent nucleic acid test result was negative, and (3) over the age of 18 years. The exclusion criteria were people with mental illness and an inability to communicate with others. Eventually, we included 34 participants. Data collection and analysis were simultaneous. Sample size was determined by data saturation, that is, no more new information emerged.

### 2.3. Data Collection

The interview outline was formulated on the basis of consulting relevant documents, policy documents and Internet information at home and abroad according to the research purpose, and made through group discussion. The interview outline was specifically tailored to each category of participants, but all focused on participants’ experiences and perceptions during the participation process, including how to organize implementation, participate in actions, and challenges encountered. The details of the interview outline is presented in Table 1.

Data was collected from August 2021 to September 2021. The researcher used a one-to-one semi-structured in-depth interview and introduced the research purposes, methods, and main contents to the participants prior to the interview, and each interview lasted 30–40 min. In order to protect the privacy of participants, the interviews were completed in a private environment, at the same time, we coded for all participants. In order to ensure the safety of researchers and participants, all participants were required to wear masks and maintain a social distance of 1.5 m during the interview.

### 2.4. Data Analysis

Within 24 h of the interview, the interview verbatim was transcribed to a transcript and then imported into the Nvivo Version 12 (Beijing Huanzhong Ruichi Technology Co., LTD, Beijing, China) for further analysis. The process for analysis followed Colaizzi’s 7-step method [22,23,24] and it was inductive thematic analysis. The specific analysis steps are shown in Table 2. In Step 1, the researcher fully familiar and understood all the content provided by the participants by repeated and careful reading of the collected information. In Step 2, the researcher conducted a word-by-word analysis of the data to identify and extract important and meaningful statements related to the research problem. In Step 3, the researcher constructed/encoded the implications of recurring perspectives. In Step 4, the researcher gathered the encoded views to look for meaningful common concepts and form the prototype of the theme. In Step 5, the researcher needed to give a detailed description of each topic generated in Step 4 and could extract and add typical original statements from the participants. In Step 6, the researcher put similar topics and their descriptions together for repeated comparisons, identified and drew out similar views, and then constructed a short and dense phrase, the theme. In order to ensure the reliability of the research results, two researchers repeatedly compared, verified and analyzed the data. If a controversial theme emerged, they would consult with the research team and then determined the final encoding and themes. The final step was to return the resulting motif structure to the participants for verification. Data was collected and analyzed in Chinese, and after the final step was confirmed by the participants, the data was translated into English and then independently backtranslated into Chinese by two researchers to ensure accuracy.

### 2.5. Rigor

During the interview, the researcher gave the participants ample time for reflection and applied interview techniques such as questioning, listening, responding, following up and repeating to encourage the participants to express their true views and feelings. At the same time, non-verbal behaviors such as tone of voice and body were observed and recorded to ensure credibility of the data. Our team has relatively extensive experience in qualitative research, in which the researchers minimized bias or guiding language. For reliability, we tried to analyze data based on Colaizzi’s 7-step method [22,23,24] and used the qualitative research analysis software Nvivo Version 12. For accuracy, the forward and backward translation processes were repeated.

### 2.6. Ethical Consideration

This study was approved by The Ethical Committees of School of Nursing of Yangzhou University (IR code: YZUHL2021016). All participants had the authority for informed consent and freedom to withdraw from the study at any time was guaranteed. Data were kept confidential. According to the interview order, the managers were numbered with English letter M (M1–M7), the health professionals were numbered with English letter H (H1–H8), the community volunteers were numbered with English letter C (C1–C6) and the residents were numbered with English letter R (R1–R14).

## 3. Results

A total of 35 participants (seven managers, eight health professionals, six community volunteers, 14 residents) were recruited. Among the managers’ participants, there were different professional title levels: staff member (*n* = 5), division chief (*n* = 1), section chief (*n* = 1). Of the eight health professionals, six (75%) were nurses and of which two (33.3%) were male. The average age of community volunteers was 35.67 years. Of the 14 residents, six (42.6%) were male, and four (28.6%) have been retired. More detail is presented in Table 3.

Three themes were revealed through the data analysis: (a) unreasonable and unsafe testing points layout settings, (b) human and medical resources challenges, and (c) potential infection risk. A detailed description of the themes, sub-themes and units of meaning can be found in Table 4.

### 3.1. Unreasonable and Unsafe Testing Points Layout Settings

#### 3.1.1. Unreasonable Layout

After the first case of COVID-19 was confirmed positive, the Yangzhou government quickly took measures to set up centralized sampling points for nucleic acid testing in communities, streets and other places. However, in the previous large-scale nucleic acid testing, participants said that the layout of some testing points lacked reasonable planning, and some community cleaning areas, buffer areas, pollution areas and other areas were not clear.

“In fact, each testing point should be divided into cleaning area, buffer area, pollution area, do you know? Places such as staff rest area, storeroom are in the clean area, where nucleic acid testing for residents is the pollution area, and then buffer area.” (M1)

“I think it should be set up conspicuous signs, clearly marked, because I found that some people did not work in the prescribed area.” (M4)

The elderly and the disabled have relatively poor physical quality, and long waiting may have bad consequences for them. However, some testing points did not take this into account and not set up a green channel for them.

“There are many old people in the queue, and there is something wrong with their legs, but they have to line up with the young people for nucleic acid testing, sometimes for one or two hours.” (C3)

“I am afraid that they cannot handle it. I heard that some old people fainted in some testing points.” (R5)

The number of people in some communities was large, but not enough testing points were established in the early stage of nucleic acid testing, they entered the same testing point for testing at the same time, resulting in a long overall testing time and high pressure on the testing points. Some participants believed that communities with more than a certain number of populations should set up testing points alone or reasonably arrange the time, arranging the residents to enter in batches.

“Too many people, the residents of several communities have come to do nucleic acid testing. Oh, there were really too many people, and we had to work from morning to night.” (H4)

“More testing points should be set up. Some communities with a large population, then we should set up separate testing points to do a good job of diversion and planning. Otherwise, on the one hand, the residents will have to wait for a long time, and on the other hand, the staff will very tired.” (M4)

#### 3.1.2. No Emergency Measures

Some participants also required the need for a safe and reliable environment. In order to better respond to emergencies, in addition to material areas, information collection areas and sampling areas, emergency areas were also needed. At least one doctor was required to be provided on site in the emergency area.

“I hope the testing points can set up emergency areas, which will make me more secure, and have better arrange several doctors for us.” (R8)

When residents have a sudden disease accident at the scene, they need to deal with them in time, and the effective treatment depends on full preparation. Some participants paid attention to this point, and they believed that they should do a good job in the emergency plans and disposal process for residents’ emergency diseases.

“There is a possibility that residents may have an accident during the testing process. Without good emergency measures, our organization is a failure and we must ensure the safety of our residents.” (M1)

### 3.2. Human and Medical Resources Challenges

#### 3.2.1. Inadequate Resources

Due to the sudden outbreak, the community was caught off guard. Although the government quickly organized large-scale nucleic acid testing initially, the actual number of staff was insufficient relative to the amount of demand due to limited access to recruit staff. There were not enough staff to guide residents sequentially through code checks, temperature measurements, registration and nucleic acid sampling.

“We were very tired, there were not enough partners to work with us.” (C4)

“I was confused, I didn’t know what to do, and I thought the organization was messy. We could only imitate the people in front of us to do the same things, because there were not enough staff to guide us.” (R7)

Sometimes, some testing points was not adequately equipped with personal protective equipment (PPE) for health professionals, such as N95 masks, protective suits and double-layer latex gloves. It was not perfect protection and could cause the fear of being infected, while those working in high-risk areas were even more afraid.

“In some testing points, the epidemic prevention materials are insufficient. Sometimes we can only wear disposable isolation clothes, and there are not enough protective clothes for us. You know, we will be afraid.” (H1)

“We are afraid of being infected. Sometimes I worry about staff who work in high-risk areas, because I have colleagues there who I think would be more at risk.” (H4)

#### 3.2.2. Irrational Deployment of Resources

During the outbreak, Yangzhou received help from all sectors of the community, and manpower and medical resources were assigned to various testing points in Yangzhou. However, participants also said that resources were not properly allocated and then caused waste. For example, the health professionals who came to support were assigned to work at different nucleic acid testing points every other day, which resulting in them having to adapt to a new environment each time and not cooperating with local staff well.

“I’m very proud that we are sent to support Yangzhou. I can contribute my strength. But what I didn’t expect is that we were sent to different testing points every new day. This means that we need to readjust every day, whether the environment or partners. I think it’s unreasonable.” (H3)

“We worked together with different health professionals each time.” (C1)

Some participants felt that better mobilization and equipping of resources were needed to provide strong support for better accomplishment of various epidemic prevention and control tasks, including nucleic acid testing.

“We were in a hurry, and we posted the fundraising information (manpower and medical resources) online...Glad that we have succeeded…If we want to beat the epidemic faster and better, we must pay attention to the details and a rational deployment of resources is essential.” (M2)

“With health professionals working on the front line, we must of course do a good job of logistics for them so that they have no worries.” (M5)

#### 3.2.3. Health Professionals Lack Standardized Training

In general, the efficiency of the testing was very high, but in order to complete the testing task within the prescribed time, it may lead to non-standard operation. Some participants felt that health professionals were not professional enough and were afraid of being infected.

“The health professional did not change their gloves between the testing of the two testers before and after, but used disinfectant water to disinfect. I don’t know if this is a standard process, I just think it’s very dangerous for me. The health professional’s gloves touched my lip. I felt like I was collapsing.” (R10)

At the same time, the health professionals may have different levels of collection methods, which may lead to failure to collect the correct specimen, resulting in false negatives for some specimens.

“I feel that some health professionals act very gentle when collecting samples, but some are not. It’s strange that some health professionals could stay for a long time while collecting samples in my throat and I felt nauseous and wanted to vomit, but when some other health professionals collected it, I didn’t feel anything, and I suspected they didn’t collect the correct specimen.” (R6)

#### 3.2.4. Health Professionals Suffer from Heatstroke

Full protective gear can give health professionals a bad feeling, including a full complement of N95 masks, protective suits, gloves, goggles and other protective measures. The airtight protective clothing combined with the fact that the outbreak occurred during the summer months required the health professionals to endure unimaginable heat, and they suffered heat stroke.

“I felt too hot and uncomfortable. We needed to wear airtight protective clothing and the masks made it hard for me to breathe. After several hours of work I was soaked with sweat. Once I was testing a resident, I felt the sky spinning and then I had heat stroke.” (H2)

“I saw a health professional get heatstroke, and I love them very much.” (C2)

### 3.3. Potential Infection Risk

#### 3.3.1. People Who Should Be Getting Nucleic Acid Testing Miss Screening

Multiple rounds of nucleic acid testing have been implemented, and residents were required to participate in each testing. However, sometimes the rate of nucleic acid testing did not reach 100%. On the one hand, some residents have little awareness of protection and lacked sufficient attention to the dangers of the Delta virus. They needed to be persuaded several times before they were willing to undergo nucleic acid testing.

“There is no doubt that some residents are not aware of the horror of this virus, they do not cooperate in participating in the nucleic acid testing and they feel safe. But if in case any of them is infected but they don’t know it, that is a very bad situation. We will do our best to screen those who have not followed the notification to do the nucleic acid testing and send someone to persuade them.” (M1)

On the other hand, older people received information more slowly compared to young and middle-aged people. Sometimes they could not receive notice to do nucleic acid testing, which lead to them missing the testing.

“Most older people have mobile phones, but they only answer calls and can’t use some of the smart features. A lot of information is posted online and the reception of information for the elderly is really a problem, so we can’t blame the elderly too much for not coming to test because they don’t know it. Right? But this is a real problem for us.” (C3)

“No one informed me to make nucleic acid testing, and I didn’t know how to do it.”(R14)

#### 3.3.2. Non-Compliance Behaviors in the Testing Queue

Due to the lack of good planning in the early stage of nucleic acid testing, residents were not notified to do testing in a certain order of testing, which led to a large number of people queuing up together to be tested. The queue for testing was long, and in the process, some residents made some bad actions that violated the rules, such as taking off their masks, spitting, etc. During multiple rounds of nucleic acid testing, several testing points caused virus transmission.

“The queue for the test was long and during this time I saw people taking off their masks and talking to others. What’s worse is that there were people spitting, which is a very dangerous thing to do. If we’re not lucky, everyone at the site will be infected and that’s bad.” (C5)

“Have you seen the news? There is a testing point where many staff have been infected along with residents, so there must be better management on site or the consequences will be unthinkable.” (M5)

#### 3.3.3. Exposed Garbage

The issue of medical waste was also brought to the attention of the participants throughout the nucleic acid testing process. As some of the testing points were set up in open locations and in case of bad thunderstorms, there would be waste seepage and spillage.

“As you can see, the conditions at our testing point are still relatively simple. It can’t be helped, for safety. It’s fine on a good day, but I’m afraid that in a thunderstorm the rubbish won’t be easy to handle.” (H6)

Some testing points did not strictly implement the medical waste disposal norms, for the recycled waste was not tied up in a timely manner. In addition, some collected medical waste was not disposed of in time, but was left in living areas for a long time.

“Is it really okay that rubbish bags filled with waste are just placed in our neighbourhood and no one is getting rid of them in time? And some of the bags are not tied up, which is my biggest concern, I hope someone will clean them up in time.” (R5)

## 4. Discussion

This research identified a series of challenges that affect the smooth development of large-scale nucleic acid testing and, the research was conducted at the early stages of the Delta virus outbreak in Yangzhou. Although the comprehensive large-scale nucleic acid testing strategy helped to contain the outbreak successfully in Yangzhou, for all those people involved in the nucleic acid testing, including managers, health professionals, community volunteers or residents, they all encountered some problems. Participants showed many concerns, and they also focused on the organization and management of the testing points and how to make sure that the testing can be carried out more efficiently.

Commercial laboratories, medical clinics, hospitals, pharmacies, schools and churches were used as testing points for nucleic acid testing in the United States [24]. To ensure that large-scale nucleic acid testing can be conducted more safely and efficiently, South Korea chose open tents and temporary buildings as testing points [6], which was similar to our research. Given the high transmissibility of the Delta virus, and in order to reduce cross-infection and reduce the risk of infection, management standards needed to be strictly formulated. Actually, cleaning zone, semi-contaminated zone and contaminated zone were strictly separated [25]. When there were special people such as the elderly or the disabled, we needed to pay more attention to the layout [26]. As a matter of fact, the sampling cells should be configured in accordance with the specification of “five groups and two channels”, including code checking, temperature measurement, sampling, registration and auxiliary groups. Green channels should be set up for independent sampling of vulnerable groups such as the over 60s and the disabled [27], to ensure all people could be tested in a safe and orderly manner. The theme of emergency measures has been reported in the study by Rong et al., who argued that managers should enhance the public incident rescue and emergency medicine capacity of health professionals in non-emergency situations [28]. It is particularly important to develop emergency plans to ensure that health professionals can effectively coordinate and regulate when responding to emergencies.

Human and medical resource management was a common challenge when responding to COVID-19, with inadequate human and material resources being a common problem [29]. It may be related to the economic capacity of the country on the one hand [30], and the high rate of infection and transmission of the virus on the other hand, with a high consumption of resources [31]. A qualitative study of the early experience of the off-site COVID-19 testing center found that staff protection was a challenge due to the lack of PPE [32]. However, from all over the world, hospitals and other health facilities have reported deficient supplies of PPE [33], staff would experience negative emotions such as anxiety, depression, and fear [34]. Working at high-risk positions was also a significant factor in the trauma of health care workers [35].

In this study, we learned that the health professionals who came from other regions to support Yangzhou did not have a fixed working place, which led to constant changes in personnel. Some studies suggested that consistent staffing on all days helped to optimize proficiency and helped in cooperation [36], while some studies addressed it through interdisciplinary collaboration and active reinforcement programs implemented [37]. Information technology should be fully used in nucleic acid testing, and it was feasible to recruit volunteers and materials through the online news platform [38].

Using the existing online platforms such as WeChat, QQ, Weibo, TikTok and other platforms to publish information on recruiting staff and raising materials (you can quickly send voice messages, videos, pictures and text through the network), to solve the problem of manpower and materials. At the same time, supportive relationships must also be established, which was one of the keys to success [39]. Neighboring communities should establish regional integration and cooperation mechanisms to complement each other in terms of epidemic prevention materials and human resources to strengthen joint prevention and control [40]. When PPE was insufficient, immediately promoted the effective and rational allocation of resources by assessing the needs of the community and minimizing the amount used for non-essential personnel. At the same time, a priority must be determined for resource allocation, allocating resources first to high-risk areas where health professionals were at a higher risk of infection.

The lack of standardized training for health professional was also a challenge reported by participants. Training was required for health professionals before they formally began to work, including nasopharyngeal swab collection and PPE use, which would increase proficiency in the operation [41]. Otherwise, the accuracy and sensitivity of the nucleic acid testing may be compromised, resulting in “false negatives”. Additionally, training was a good way to avoid the mistake of health professionals’ gloves touching the residents’ lips [42]. In the study by Goldberg et al., health professionals changed gloves when testing new residents [41], but it was feasible to use only alcohol for routine disinfection and not change gloves [6]. The lack of standardized training not only placed a burden on residents during testing, but also on the health of health professionals [43].

In this study, we also learned that health professionals have suffered considerable physiological challenges at work. The full set of PPE was a big burden on them, and health professionals were at risk of heat stroke, and the same theme has been reported in some studies [44,45]. This required the people in charge of testing points to make efforts to improve the health security of health professionals, and to ensure adequate water, food, air conditioning equipment and rest places [46]. When testing points were set up outdoors and cooling equipment such as air conditioning cannot be applied, ice could be provided to cool them down. Reasonably arrange the working hours of professional health personnel with regular rotation, as far as possible to reduce the physical discomfort caused by the long time of work.

Spring believed that this era was difficult and uncertain in the context of the pandemic, and good health literacy has never been more crucial to survival [47]. Every resident who was asked to be tested was expected to take the initiative to participate, but we have found that not everyone had a good sense of protection and health awareness. The study found that some residents resisted or did not take the test at all, and that some residents made some bad actions that violated the rules while waiting for the test, such as not keeping social distancing of about 1.5 m or not wearing a mask [48]. Attention was on the elderly group who, despite being asked to participate in each round of nucleic acid testing, faced significant barriers in accessing information [49]. Due to the sharing of COVID-19-related information to the public through new media, the lack of access to information has caused the elderly to not realize the severity of COVID-19 and have a weak awareness of self-protection [50]. At the same time, notifications of being tested were usually made through chat software, which was not available to older people in a timely manner, resulting in them missing out on nucleic acid testing [51]. These behaviors were not only harmful to individuals, but also to society [47].

Therefore, it is important to introduce relevant codes of conduct and to strengthen and broaden the channels of communication to ensure that all people are aware of the need for nucleic acid testing and the dangers of the virus, and that the government plays an important role in this regard [52]. Those who do not participate in nucleic acid testing as required will be listed as trust-breaking personnel and listed on the information platforms, which will affect their employment, appointment, loans and other behaviors in the future. In addition, community-based grid management can grasp the information of the floating population at any time, ensure accurate mapping, effectively increase the participation rate in nucleic acid testing, and achieve the goal of full inspection. Therefore, it can be widely used to curb the spread of the virus in the future [40].

The storage of medical waste was of great concern. Since the outbreak of the coronavirus disease, medical protective equipment such as masks, protective clothing and gloves have been consumed greatly, and the amount of harmful waste has been terrible. Medical waste was a vector for SARS-CoV-2, therefore, proper disposal of COVID-waste was therefore immediately required to lower the threat of pandemic spread, and also reduce the threat to public health and the social environment [53]. COVID-waste was required to be stored for no more than 24 h, classification of waste was the first step for the management of COVID-waste [54]. Additionally, COVID-waste must be stored in double special yellow bags, tightly sealed and collected, and stored separately by labelling it “COVID-19 waste”. At the outset, a bin with a lid should be selected as the first barrier. Once the bags were filled, they should be sprayed with disinfectant spray on the inside and outside surfaces by a dedicated logistician and tied up in time to send to a central collection point, and await transfer by a dedicated vehicle, avoiding being left in the sampling area for long periods of time. At the same time, the environment, air, and ground around the trash bin should be sprayed with a 2000 mg/L chlorine-containing disinfectant [55].

The outbreak in Yangzhou has served as a good warning. To avoid being unprepared in the event of an epidemic, the local government can organize the community to participate in the drills to improve the level of knowledge and preparedness of the community residents [56]. Through the drill, residents can be familiar with the whole appointment registration and testing process of nucleic acid testing, laying a solid foundation for the rapid participation in large-scale nucleic acid testing for the entire population in times of emergency. In this process, we should focus on the elderly group, the role of family support should be given full play, as this is the core part of the social network of the elderly, and it is more reliable to get information through the mouths of their children and grandchildren [57]. Children and grandchildren should shoulder part of the responsibilities of regulating the behavior of the elderly, including raising awareness of epidemic prevention, not gathering, and participating in large-scale nucleic acid tests as instructed. Given the current spread of the epidemic and the effectiveness of large-scale nucleic acid testing in contain the virus, our findings may provide lessons for other countries around the world to implement more efficient and rational large-scale nucleic acid testing.

This study was conducted only for Yangzhou, a third-tier city in China, which limits the representation of the samples, while the results may not be well generalizable.

## 5. Conclusions

This study identifies the challenges suffered by different populations during large-scale nucleic acid testing. Although nucleic acid testing has been applied on a large scale in China during the COVID-19 epidemic in 2019–2020, the sudden outbreak of the more dangerous the Delta variant (B.1.617.2) has created greater and higher requirements on large-scale nucleic acid testing, making positive planning and adequate preparation are an important part of the smooth development of testing. Successful large-scale nucleic acid testing relies on the cooperation and efforts of all to support containment of the spread of the virus. Although there are relevant health policies to guide large-scale nucleic acid testing, there is still room for modification. Health institutions should, on the basis of the previous policies, have stricter requirements on the layout of testing points and the management of manpower and material resources, and should also pay attention to the factors causing the potential infection risk, and formulate health policies in line with their own regional conditions.

## Figures and Tables

**Table 1 ijerph-19-01573-t001:** Samples of interview question.

Participants	Questions
Managers	Please tell me about your experiences in the organization of the large-scale nucleic acid testing? What is your deepest feeling during the process?
	What challenges have you experienced in the organization of the large-scale nucleic acid testing?
	How do you meet these challenges?
	What do you think needs to improve during the large-scale nucleic acid testing?
	What is your expectation of support to help you organize the large-scale nucleic acid testing better?
Staff members (Health professionals and Community volunteers)	Please tell me about your experiences in working during the large-scale nucleic acid testing? What is your deepest experience during the process?
	What challenges have you experienced in working during the large-scale nucleic acid testing?
	What protections did you take to avoid getting infection?
	Did you get some welfare security from the community or the testing points, such as diet, rest or pay?
	What is your expectation of support to help you work better in the large-scale nucleic acid testing?
Residents	Please tell me about your experiences in participating the large-scale nucleic acid testing?
	Did you participate in every nucleic acid testing as required?
	What challenges have you experienced in participating during the large-scale nucleic acid testing?
	What is your expectation of support to help you more willing to take part in the large-scale nucleic acid testing?

**Table 2 ijerph-19-01573-t002:** Data analysis steps.

	Specific Contents
Step 1	Read the source carefully
Step 2	Extract phrases or sentences related to the research phenomenon
Step 3	Encode important statements that occur repeatedly
Step 4	Categorize the codes and integrate the results
Step 5	Provide a detailed description of the research phenomenon
Step 6	Reduce the detailed description to form the structural framework
Step 7	Return the research object for verification

**Table 3 ijerph-19-01573-t003:** Participants’ characteristics.

	Age (Year)	Gender	Education	Job Status
Managers (7)	30–39: 1 40–49: 3 >50: 3	Male: 5 Female: 2	Bachelor: 4 Master: 3	Section member: 5 Division chief: 1 Section chief: 1
Health professionals (8)	<30: 4 30–39: 3 40–49: 1	Male: 2 Female:6	Bachelor: 6 Master: 2	Doctors: 2 Nurses: 6
Community volunteers (6)	<30: 1 30–39: 3 40–49: 1 >50: 1	Male: 2 Female: 4	Primary: 2 High school: 2 University: 2	jobless: 3 Worker: 2 Retired: 1
Residents (14)	<30: 3 30–39: 2 40–49: 4 50–59: 2 >60: 3	Male: 6 Female: 8	Primary: 3 High school: 5 University: 6	jobless: 3 Worker: 7 Retired: 4

**Table 4 ijerph-19-01573-t004:** Themes, sub-themes and units of meaning.

Themes	Sub-Themes	Units of Meaning
Unreasonable and unsafe testing points layout settings	Unreasonable layout	“In fact, each testing point should be divided into cleaning area, buffer area, pollution area, do you know? Places such as staff rest area, storeroom are in the clean area, where nucleic acid testing for residents is the pollution area, and then buffer area.” (M1)
		“I think it should be set up conspicuous signs, clearly marked, because I found that some people did not work in the prescribed area.” (M4)
		“There are many old people in the queue, and there is something wrong with their legs, but they have to line up with the young people for nucleic acid testing, sometimes for one or two hours.” (C3)
		“I am afraid that they cannot handle it. I heard that some old people fainted in some testing points.” (R5)
		“Too many people, the residents of several communities have come to do nucleic acid testing. Oh, there were really too many people, and we had to work from morning to night.” (H4)
		“More testing points should be set up. Some communities with a large population, then we should set up separate testing points to do a good job of diversion and planning. Otherwise, on the one hand, the residents will have to wait for a long time, and on the other hand, the staff will very tired.” (M4)
	No emergency measures	“I hope the testing points can set up emergency areas, which will make me more secure, and have better arrange several doctors for us.” (R8)
		“There is a possibility that residents may have an accident during the testing process. Without good emergency measures, our organization is a failure and we must ensure the safety of our residents.” (M1)
Human and medical resources challenges	Inadequate resources	“We were very tired, there were not enough partners to work with us.” (C4)
		“I was confused, I didn’t know what to do, and I thought the organization was messy. We could only imitate the people in front of us to do the same things, because there were not enough staff to guide us.” (R7)
		“In some testing points, the epidemic prevention materials are insufficient. Sometimes we can only wear disposable isolation clothes, and there are not enough protective clothes for us. You know, we will be afraid.” (H1)
		“We are afraid of being infected. Sometimes I worry about staff who work in high-risk areas, because I have colleagues there who I think would be more at risk.” (H4)
	Irrational deployment of resources	“I’m very proud that we are sent to support Yangzhou. I can contribute my strength. But what I didn’t expect is that we were sent to different testing points every new day. This means that we need to readjust every day, whether the environment or partners. I think it’s unreasonable.” (H3)
		“We worked together with different health professionals each time.” (C1)
		“We were in a hurry, and we posted the fundraising information (manpower and medical resources) online...Glad that we have succeeded…If we want to beat the epidemic faster and better, we must pay attention to the details and a rational deployment of resources is essential.” (M2)
		“With health professionals working on the front line, we must of course do a good job of logistics for them so that they have no worries.” (M5)
	Health professionals lack standardized training	“The health professional did not change their gloves between the testing of the two testers before and after, but used disinfectant water to disinfect. I don’t know if this is a standard process, I just think it’s very dangerous for me. The health professional’s gloves touched my lip. I felt like I was collapsing.” (R10)
		“I feel that some health professionals act very gentle when collecting samples, but some are not. It’s strange that some health professionals could stay for a long time while collecting samples in my throat and I felt nauseous and wanted to vomit, but when some other health professionals collected it, I didn’t feel anything, and I suspected they didn’t collect the correct specimen.” (R6)
	Health professionals suffer from heatstroke	“I felt too hot and uncomfortable. We needed to wear airtight protective clothing and the masks made it hard for me to breathe. After several hours of work I was soaked with sweat. Once I was testing a resident, I felt the sky spinning and then I had heat stroke.” (H2)
		“I saw a health professional get heatstroke, and I love them very much.” (C2)
Potential infection risk	People who should be getting nucleic acid testing miss screening	“There is no doubt that some residents are not aware of the horror of this virus, they do not cooperate in participating in the nucleic acid testing and they feel safe. But if in case any of them is infected but they don’t know it, that is a very bad situation. We will do our best to screen those who have not followed the notification to do the nucleic acid testing and send someone to persuade them.” (M1)
		“Most older people have mobile phones, but they only answer calls and can’t use some of the smart features. A lot of information is posted online and the reception of information for the elderly is really a problem, so we can’t blame the elderly too much for not coming to test because they don’t know it. Right? But this is a real problem for us.” (C3)
		“No one informed me to make nucleic acid testing, and I didn’t know how to do it.” (R14)
	Non-compliancebehaviors in the testing queue	“The queue for the test was long and during this time I saw people taking off their masks and talking to others. What’s worse is that there were people spitting, which is a very dangerous thing to do. If we’re not lucky, everyone at the site will be infected and that’s bad.” (C5)
		“Have you seen the news? There is a testing point where many staff have been infected along with residents, so there must be better management on site or the consequences will be unthinkable.” (M5)
	Exposed garbage	“As you can see, the conditions at our testing point are still relatively simple. It can’t be helped, for safety. It’s fine on a good day, but I’m afraid that in a thunderstorm the rubbish won’t be easy to handle.” (H6)
		“Is it really okay that rubbish bags filled with waste are just placed in our neighbourhood and no one is getting rid of them in time? And some of the bags are not tied up, which is my biggest concern, I hope someone will clean them up in time.” (R5)

## Data Availability

The data presented in this study are available on request from the corresponding author.

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
