# Peer review of "Challenges Faced in Large-Scale Nucleic Acid Testing during the Sudden Outbreak of the B.1.617.2 (Delta)"

_ijerph, 2022, doi:10.3390/ijerph19031573_

Round 1

Reviewer 1 Report

After carefully review this paper, “Challenges faced in large-scale nucleic acid testing during the sudden outbreak of the B.1.617.2 (Delta)”, the following comments are listed for your reference:

  1. Introduction (P1, L28-32): I would suggest to include further references to support your shorter sentences.
  2. Introduction (P1-2, L27-75): This section can be difficult to read at times due to the overuse of short sentences, however longer sentences could improve its readability. Please review linkers used throughout the manuscript. I would suggest to use alternative linkers rather than only “however” or “therefore” (they are repeated quite often).
  3. Methods (P2, L86-87): Please clarify “sample size was determined by data saturation”. Was data collection and analysis simultaneous? Did data analysis begin when all data had been collected? If the authors used a phenomenological study, how was bracketing, for example, woven in the analysis? Was analysis an iterative process? What was the approach to coding?
  4. Methods (P2-3, L88-107): Was data collected and analysed in English or Chinese? And, if data was analysed in English, how did the author(s) ensure that the themes were representative of what participants have said?
  5. Methods (P2, L92-93): I would exchange the order of “participants’ perceptions and experiences” to “participants’ experiences and perceptions” as our perceptions are supposed to be based on experiences.
  6. Methods (P3, L98-100): Data collection took place from August to September 2021. Did authors had to follow any particular safety measure to conduct their face-to-face interviews? Further rationale would help readers to better understand the data collection process.
  7. Methods (P3, L112-114): Authors refer to “themes”, but I could not find any reference to this type of analysis elsewhere. Was it inductive thematic analysis? What was the approach to coding? Please clarify this section.
  8. Results (P3, L128): Authors claim “2 (25%) were male and 6 (75%) were nurses”. Do you mean “female”?
  9. Results (P4-7, L134-297): I would recommend to separate paragraphs and quotes to make it more easy-reading.
  10. Results (P4-7, L134-297): Participants are assigned letters and numbers. Please include this information into the methods section (data collection): e.g., “All transcripts were anonymized, using the letter “M” (managers), “H” (health professionals), “C” (community volunteers) and “R” (residents) followed by the participant number.
  11. Results (P5, L189): One participant (C4) says “there were not enough partners to work with us”. Do they mean “fellow workers”?
  12. Discussion (P8, 343-346): Did this sentence emerge from your results? Please clarify or add a reference to support your statement.
  13. Discussion (P8, 346-347): Please clarify “supportive relationship must also be established”. Do you mean among professionals? From managers to personnel? From society to healthcare workers?
  14. Discussion (P10, L425-426): Were participants treated as a homogenous group? If so, please include this information into your limitation section. I would suggest to include further lines of action derived from your results (e.g., the long-term implication for vulnerable populations).
  15. Others: Please review some typo errors (e.g., L110 “analysis[20]”, L130 “Table2” or L194 “equipment(PPE)”).
  16. References (P10-12, L455-565): References need attention. This section should be reviewed following author guidelines.
  17. English: Additional proof reading would enhance it greatly. Some sentences could be restructured for clarity.

Reviewer 2 Report

Attached the comments for the authors

Round 2

Reviewer 1 Report

Thank you for sending your revised paper entitled “Challenges faced in large-scale nucleic acid testing during the sudden outbreak of the B.1.617.2 (Delta)” to IJERPH and take my recommendations into account. Just few minor revisions to consider:

  1. Methods (P3, L133): If an inductive analysis was used, please include this information within your manuscript.
  2. Results (P4, L209): I would suggest to clarify this sentence as it can be misleading. Did you mean “there were 6 nurses and of which 2 were male”?
  3. References: Please review typo errors within your references in text (e.g., “variants[16]” should be “variants [16]”).
